# micronuclAI enables automated quantification of micronuclei for assessment of chromosomal instability
Miguel A. Ibarra-Arellano [1,11], Lindsay A. Caprio [2,3,11], Aroj Hada[1,4], Niklas Stotzem[1,5,6,7], Luke L. Cai[2,3], Shivem B. Shah [2,3], Zachary H. Walsh[2,3], Johannes C. Melms [2,3], Florian Wünneman[1], Kresimir Bestak[1], Ibrahim Mansaray[1], Benjamin Izar [2,3,8] & Denis Schapiro [1,4,9,10]

Chromosomal instability (CIN) is a hallmark of cancer that drives metastasis, immune evasion and treatment resistance. CIN may result from chromosome mis-segregation errors and excessive chromatin is frequently packaged in micronuclei (MN), which can be enumerated to quantify CIN. The assessment of CIN remains a predominantly manual and time-consuming task. Here, we present micronuclAI, a pipeline for automated and reliable quantification of MN of varying size and morphology in cells stained only for DNA. micronuclAI can achieve close to human-level performance on various human and murine cancer cell line datasets. The pipeline achieved a Pearson's correlation of 0.9278 on images obtained at 10X magnification. We tested the approach in otherwise isogenic cell lines in which we genetically dialed up or down CIN rates, and on several publicly available image datasets where we achieved a Pearson's correlation of 0.9620. Given the increasing interest in developing therapies for CIN-driven cancers, this method provides an important, scalable, and rapid approach to quantifying CIN on images that are routinely obtained for research purposes. We release a GUI-implementation for easy access and utilization of the pipeline.

Chromosomal instability (CIN) is a hallmark of aggressive cancers[1–3], and developmental and age-related disorders[4]. In cancer, CIN drives tumor progression, heterogeneity, immune evasion, and treatment resistance across a range of tumor lineages[3,5–10]. CIN may arise from different mutagens (e.g., radiation, mitotic toxins), defects in DNA repair, and most frequently, from errors in chromosome segregation during anaphase[11–13]. Following asymmetric distribution of chromatin, cells receiving excessive material frequently package DNA in micronuclei (MN). MN are variable in size (ranging from small (0.5–1 μM) to large (10–15 μm)), however less than 1/3 of the size of the primary nucleus, exist in different morphologies and locations in relation to the nucleus, and lack the normal nuclear envelope, which results in frequent rupture and release of double-stranded DNA (dsDNA) to the cytosol[14,15]. This in turn triggers the cytosolic DNA sensing machinery via cGAS-STING, that may result in production of pro-immunogenic cytokines (e.g., type I interferons) when stimulated briefly, but suppresses cytokine production and promotes STING-dependent pro-metastatic pathways when activated tonically, such as in the case of most CIN-driven cancers[7,16,17]. Thus, enumerating MN of varying qualities is a useful approach for quantifying CIN and has important functional implications for tumor behavior, immune responses, and treatment outcomes.

Several approaches, such as cytogenetics, quantitative imaging and single-cell genomics can be used for the assessment of CIN[18]. Among them, quantitative imaging via microscopy is widely used for its simplicity, low-cost, and scalability. Here, assessment of CIN is performed through the quantification of its associated structural biomarkers, including MN, anaphase bridges, and Nuclear Buds (NBUDs), which are MN that may

[1]Institute for Computational Biomedicine, Heidelberg University, Faculty of Medicine, Heidelberg University Hospital, Heidelberg, Germany. [2]Department of Medicine, Division of Hematology/Oncology, and Herbert Irving Comprehensive Cancer Center, Columbia University Irving Medical Center, Columbia University Vagelos College of Physician and Surgeons, New York, NY, USA. [3]Columbia Center for Translational Immunology, Columbia University Irving Medical Center, New York, NY, USA. [4]AI-Health Innovation Cluster, Heidelberg, Germany. [5]School of Computation, Information and Technology, Technical University of Munich, Garching, Germany. [6]Institute of AI for Health, Helmholtz Munich, Neuherberg, Germany. [7]Helmholtz Pioneer Campus, Helmholtz Munich, Neuherberg, Germany. [8]Department of Systems Biology, Program for Mathematical Genomics, Columbia University, New York, NY, USA. [9]Institute of Pathology, Heidelberg University Hospital, Heidelberg, Germany. [10]Translational Spatial Profiling Center (TSPC), Heidelberg, Germany. [11]These authors contributed equally: Miguel A. Ibarra-Arellano, Lindsay A. Caprio. ✉ e-mail: bi2175@cumc.columbia.edu; Denis.Schapiro@uni-heidelberg.de

maintain a visible stalk of nucleoplasmic material[19], among others[12,20–22]. Although quantification of these functional CIN surrogates from microscopy images is a routine process in cancer research, it is typically achieved through manual scoring. Therefore, microscopy images are divided into high power fields of view (FOV), and the total number of nuclei as well as the number of MN are counted within each FOV. The rate of MN and nuclei is then estimated over the whole image in terms of ratio between the number of MN and the number of nuclei averaged over all the FOV regions. During a standard analysis, multiple images are scored per sample, resulting often in at least 30 FOVs consisting of approximately 1000-1500 primary nuclei. Subsequently, all these images must be manually counted and analyzed[7,23–25].

The manual nature of this workflow makes it tedious, time-consuming and error prone, thus, limiting scalability. Moreover, the morphology and localization of MN represent an important source of misclassification errors. The complexity of the task is further exacerbated by the lack of a standard protocol leading to inter-observer variability in MN counting. Additionally, density of nuclei in each image and the resulting crowding makes it challenging to confidently separate and accurately assign MN events to nuclei for both humans and automated methods[26]. Thus, improved methods for automated CIN quantification are necessary to increase speed, accuracy and robustness of CIN-related research.

In this study, we aim to develop a robust and scalable solution for automatically quantifying CIN via MN counting. micronuclAI quantifies CIN using nuclei-stained images to assess the number of MN associated with each nucleus individually. We compare micronuclAI results to manual expert annotation on human and murine cell lines to assess the performance and robustness across multiple datasets and microscopy conditions. Additionally, micronuclAI offers tools for manually annotating new datasets with the PySimpleGUI and VR labeling tools. MicronuclAI is publicly available as an open-source Python package, a Nextflow pipeline, and a user-friendly web-based Streamlit application.

## Results

### Design and implementation of micronuclAI

We developed micronuclAI, a deep-learning-based pipeline to assess CIN through automated quantification of MN from nuclei-stained images. micronuclAI distinguishes itself from previous methods[26–32] as it 1) can quantify for both MN and NBUDs; 2) requires only nuclear (DNA) staining; 3) can work with 10× to 20× image objectives; 4) can work with any segmentation mask, and most importantly 5) is extensively evaluated in multiple cell lines and thus, ready for use by the community for nuclear-stained images of cell lines (Fig. 1 and Table 1). Additionally, compared to manual scoring, micronuclAI eliminates inter-observer variability and significantly reduces the CIN scoring time (Supplementary Fig. 1).

### Generation of the training data set and model selection

Expert annotation (the process of manually defining the amount of MN) was performed using a custom tool, written in Python[33], on 23 images with a resolution of 8829 × 9072 pixels of the human melanoma cell line A375. To perform the nuclear segmentation of the training images, we tested multiple methods and models: DeepCell Mesmer / DeepCell nuclear[34], Cellpose[35], and Stardist[36]. We visually inspected the generated nuclear segmentation masks and chose Stardist which generated the most accurate masks through qualitative comparison of several ROIs within the input image.

Nuclear segmentation was performed using the Stardist segmentation method with default parameters. Importantly, micronuclAI performance is robust across multiple segmentation approaches (Supplementary Fig. 2), thus the choice of the segmentation method was not critical. With the help of the segmentation masks, we isolated 84,286 nuclei. From each isolated nucleus, the number of MN present was manually counted for CIN estimation and henceforth termed as "CIN count" (Fig. 2a) (See Methods). In all analyzes, NBUDs are considered MN of distinct localization and morphology. From the labeled nuclei, 77,733 (92.23%) had a CIN count of 0; while 6553 (7.77%) of the labeled nuclei had a CIN count of at least 1. To better handle this data imbalance,

we removed 23 (0.0272%) outliers with a CIN count ≥ 4, and randomly sampled the same number of nuclei with a CIN count of 0 to the nuclei with a CIN count > 0. The balanced training dataset consisted of 12,304 nuclei from 21 of the 23 labeled images; from which 6152 (50%) had CIN count of 0; 5473 (42.74%) had a CIN count of 1; 564 (4.58%) had a CIN count of 2; and 92 (0.75%) had CIN count of 3 (Fig. 2b). From this balanced dataset of 12,304 nuclei, (90%) were used for training and (10%) used for validation. The test dataset consisted of 804 nuclei obtained from the remaining 2 hold out test images (Fig. 3a).

We selected the EfficientNet-V2-S[37] model architecture as the final model based on the average 10-fold cross-validation (CV) performance. The consistent performance across all the "folds" validated this decision, ensuring that our model is both robust and well-generalized. Furthermore, we compared the performance of the model using both a balanced and an imbalanced dataset. This comparison highlighted better model performance when trained on the balanced dataset (Supplementary Table 1).

### micronuclAI accurately identifies CIN in the hold-out test set

On the hold-out test set comprising 804 isolated nuclei, the best model (EfficientNet V2-S) achieved an average F1-weighted score of 0.9301, and a MCC (Mathew's correlation coefficient) of 0.8751 (Fig. 3b). These metrics showcase the ability of the model to accurately quantify the number of MN in each isolated nuclei image. Additionally, the attention maps correlate with the areas where MN are present, and in cases where there is no MN, the attention is focused across the entire nucleus (Fig. 3c). This shows the model has learned to identify the specific features and patterns associated with MN and conversely accurately identifies cells without MN.

### micronuclAI performance generalizes across biological and technical contexts

To assess the generalizability of micronuclAI to a new cell line, we evaluated the complete micronuclAI pipeline on images taken from the NCI-H358 non-small cell lung cancer (NSCLC) line. The rate of CIN was modulated using previously established genetic constructs. In brief, overexpression of a dominant-negative mutant of mitotic-centromere associated kinesin (dnMCAK) enhances CIN in otherwise CIN-low cell lines[38]. Thus, in addition to the parental cell line (H358) and expression of a vector control (H358-GFP), we also examined a CIN-modified derivative (H358-dnMCAK).

Evaluation of the complete pipeline was done based on the calculation of the CIN score, as a ratio of the CIN counts to the number of total nuclei present, compared between manual counts from experts and the automatic counts obtained from micronuclAI. The CIN scores for individual images were found to be very accurate for each sample resulting in an RMSE value of 0.0041, $R^2$ of 0.881, and a Pearson's correlation of 0.932. These metrics signify that the manual and micronuclAI automatic counts show a high correlation. Furthermore, the metrics improve when the CIN score is calculated as an average of multiple technical or biological replicate images, which is the standard procedure in manual counting. When taking average counts for the three genetic variants of H358 cell line; H358 control, H358-GFP, and H358-dnMCAK, the values for RMSE decreased to 0.0020, the $R^2$ increased to 0.9794, and the Pearson's correlation increased to 0.989 (Fig. 4a).

### micronuclAI performance is robust to different species, magnification and technical variation

We tested micronuclAI's generalizability to unseen nuclear stains and imaging conditions using independent datasets. Therefore, we initially leveraged the BBBC039v1[39] data which is a Hoechst-stained U2OS osteosarcoma cell line imaged at 20×. Comparing the CIN score between human expert count and our pipeline revealed that micronuclAI was able to accurately quantify the number of cells and micronuclei in the images. For 20 randomly selected images present in the dataset, micronuclAI was able to achieve an RMSE of 0.0159, an $R^2$ of 0.905, and a Pearson's correlation of 0.951 confirming the magnification invariability in images ranging from

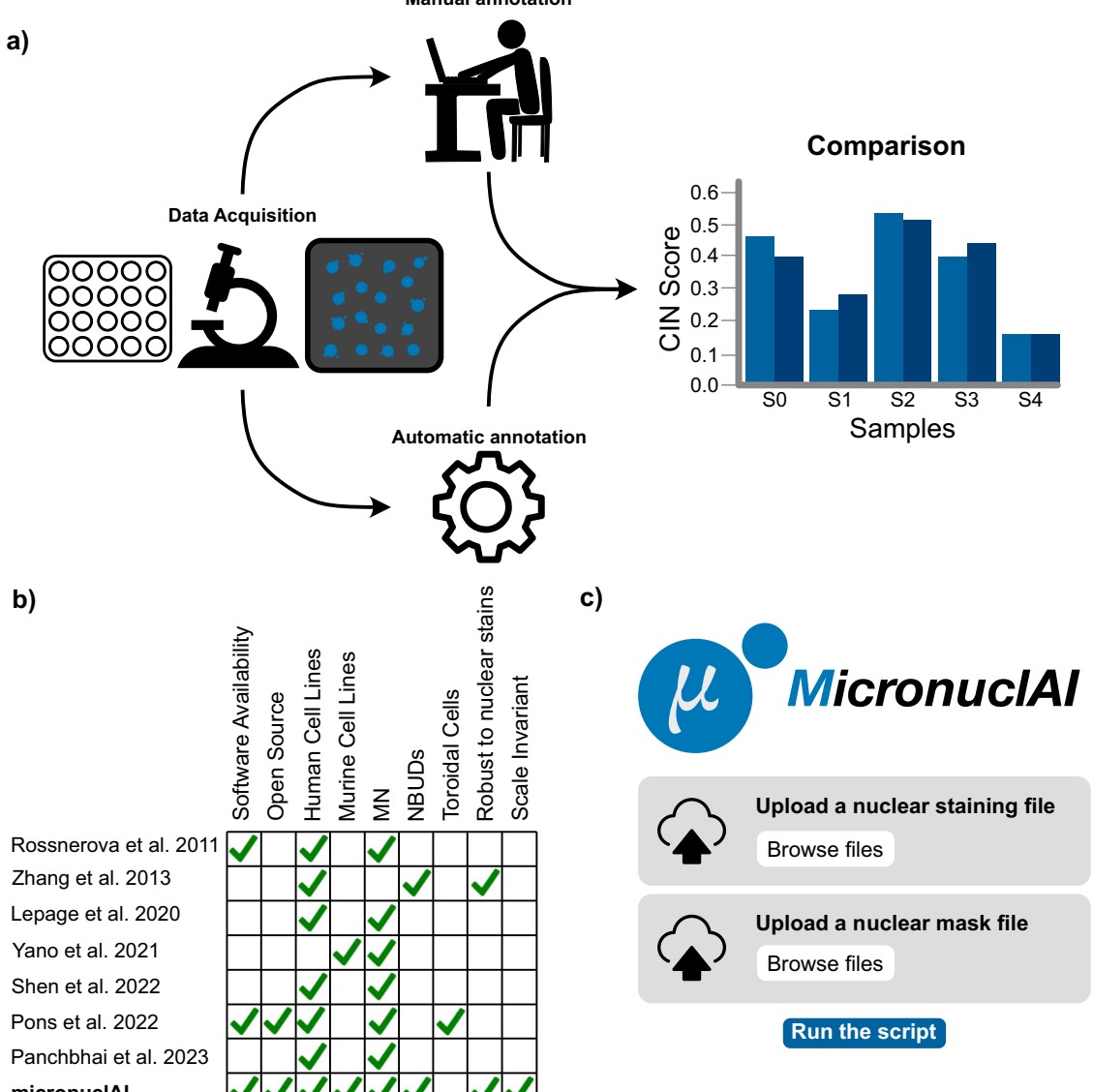

**Fig. 1 | micronuclAI pipeline overview and previous work. a** micronuclAI aims to be the go-to tool for CIN assessment via automatic quantification of MN in nuclei-stained images over the manual workflow. Evaluation of the pipeline was performed by comparing the CIN score, a ratio of the total number of micronuclei and associated structures to the total number of nuclei present, both manually and automatically in multiple cell-lines. **b** Comparison between previous methods and micronuclAI. Prior efforts to automatically quantify CIN bring their own advantages but also have certain limitations involving ability to account for multiple species, multiple MN structures (MN or NBUDs), limited software availability, and dependence on image acquisition parameters. **c** Mock-up of the micronuclAI web application (https://micronuclai.streamlit.app/).

10× to 20× magnification and stain invariability between DAPI and Hoechst stains (Fig. 4b).

Additionally, we evaluated the performance of the pipeline on murine lung cancer cell lines KP ($Kras^{G12D}Tp53^{-/-}$) ($n = 5$) and KL ($Kras^{G12D}Lkb1^{-/-}$) ($n = 5$). We observed a great degree of correlation between the results with an RMSE of 0.00684, an $R^2$ of 0.9194, and a Pearson's correlation coefficient of 0.9620. Thus, confirming the robustness of the pipeline to non-human cell lines. Importantly, we used our previously identified default parameters for preprocessing and predictions of these cell lines, thus, further optimization would likely improve the results (Fig. 4c). All the results obtained on the different cell types have been summarized in Table 2.

## Discussion

In this study, we present micronuclAI, a framework that harnesses the power of deep-learning technology and computer vision techniques for CIN assessment in cancer cell lines in situ. By focusing on MN quantification, micronuclAI offers a reliable proxy for assessing CIN at scale, providing a standardized and efficient solution.

Prior efforts recognized the importance of automating MN quantification[40,41], typically through combinations of traditional computer vision, machine learning or deep learning approaches with their own advantages and limitations as shown in Fig. 1 and Table 1. A large number of previous works have used the Cytokinesis-block MicroNucleus (CBMN) assay following the protocols defined by the HUMN (HUman Micro-Nucleus) project[42] to quantify MN within binucleated lymphocytes. Briefly, the CBMN assay is a standardized method for genotoxicity studies where biomarkers of CIN are scored in binucleated cells after cytokinesis is stopped by addition of cytochalasin B[40,43]. Other exemplary methods automate MN quantification in various other cell/tissue types after genotoxic exposures, citing no difference in results between the in vitro MN tests with or without the use of cytokinesis blockers like cytochalasin B[44,45]. Meanwhile, other methods have proposed quantification of nuclear budding MN

**Table 1 | Relevant work**

| Approach | Targeted Structures | Cell/Tissue Type | Method | Staining Requirement | Issues/Challenges |
|---|---|---|---|---|---|
| Metafer MNScore[27] | Micronuclei | Binucleated lymphocytes | Not specified | Nuclear + Membrane | - Specific for MN only |
| Zhang et al.[28] | NBUDs | Mono + Binucleated Lymphocytes | CV based (Ellipse fitting, Moving sticks, and Top Hat) | Nuclear | - Specific for NBUDs only |
| ScQuantIM[29] | Micronuclei | Colorectal cancer cell lines | CV based pipeline | Nuclear + Membrane | - Manual parameter tuning required |
| CADMDI[26] | Micronuclei | Mice Tissue | Segmentation + CV based pipeline | Nuclear + Membrane | - Manual parameter tuning required |
| Shen et al.[30] | Micronuclei | Binucleated lymphocytes | CNN for classification of nuclei type, CV based pipeline + K-means for MN quantification | Nuclear + Membrane | - High false positives (26.2%) - Requires input image to be Giemsa stained |
| PIQUE[31] | Micronuclei, Toroidal nuclei | U2OS & HeLa cells | CV based pipeline | Nuclear | - Does not quantify NBUDs - Images evaluated only at 40× |
| Panchbhai et al.[32] | Micronuclei | Binucleated lymphocytes | Faster-RCNN for object detection | Nuclear | - Only for 100× images - Cannot quantify NBUDs - Limited dataset |
| micronuclAI | Micronuclei NBUDs | Multiple Human + Mice cell lines | CV + CNN | Nuclear | |

specifically[28], or focus on a combination of MN and toroidal nuclei[31] to quantify CIN. Importantly, to our knowledge there have been no published methods for automatic quantification of MN of varying morphology, localization, across species and lineages in cancer cell lines. In addition, most methods evaluate their performance on a limited dataset plagued by inter-observer variability. Therefore, existing methods face certain limitations such as i) quantification specifically of micronuclei alone within binucleated lymphocytes, ii) quantification of NBUDs alone, iii) requirement of cellular/cytoplasmic staining in addition to nuclear staining, iv) lack of adequate evaluation, and v) are not open-source or easily available. In contrast, micronuclAI performs well across multiple biological, perturbational, and technical conditions, and achieves human-level performance at a fraction of the time, taking approximately 10 s (on a MacBook Pro M1) to score 3000 cells compared to the ~120 min for manual scoring. Its robustness extends to handling multiple nuclear staining, as demonstrated with DAPI and Hoechst datasets. Furthermore, micronuclAI proves its magnification invariance by excelling under varying microscopy conditions, including 10×, and 20× objective datasets. Furthermore, our pipeline is also robust against out-of-focus nuclei images which are common in microscopy images as the model has been trained with a limited number of blurry nuclei (Supplementary Table 2). The pipeline also outputs a percentage of out-of-focus nuclei images from the given image as a form of quality check.

Despite all its strength, micronuclAI has also limitations. First, it cannot explicitly communicate the reasoning behind its predictions, but its value lies in its consistent and scalable detection of micronuclei as a reliable indicator of CIN. We partly address this issue by using saliency maps to highlight the regions in the image that contributed the most to drive the prediction. Another limitation is the reliance on a nuclear segmentation mask. Although precise identification of the nuclear boundaries is not required and the method is robust to multiple segmentation methods, instances of extensive mis-, over-, and under-segmentation may affect the predicting capabilities of the framework so the segmentation quality cannot be fully omitted. Cases where the model does not perform well are mostly attributed to instances in images with overlapping nuclei (due to cell lines which tend to grow on-top of one another), irregularly shaped nuclei, or the presence of apoptotic cell fragments in the periphery of isolated nuclei (Supplementary Fig. 3). Instances of the same MN on two different isolated nuclei images might be present, leading to a potential double counting of the same structures within two different patches. Such instances occur on areas of the image with high nuclear/cell density that might yield misleading or inflated results. Although rare, these instances can be minimized by following two approaches: 1) changing the FOV used for the nuclei isolation step and, 2) addressing the experimental design before imaging. To quantify the frequency of these occurrences, we calculated the Intersection over Union (IoU) between all the patches in an image. 0.00168% of the pairs had an IoU ≥ 0.5 signifying that only a negligible amount of overlap is present in the dataset, hence we can safely disregard the concern for double counting—especially when also controlling for cell density. Lastly, we would like to note, that MN enumeration from H&E staining is currently not supported due to a lack of established ground-truth data obtained from this staining modality.

In summary, micronuclAI is a robust and scalable tool for quantifying CIN, and will be a critical tool for understanding CIN biology and its role in tumor biology and treatment responses.

## Methods
### Cell culture
A375 and NCI-H358 were obtained from the American Type Culture Collection (ATCC). KP ($Kras^{G12D}Tp53^{-/-}$) and KL ($Kras^{G12D}Stk11^{-/-}$) are descriptors of a syngeneic pair of murine non-small cell lung cancer lines and were kindly gifted from Dr. Kwok-Kin Wong. KP is characterized by both oncogenic mutations in *Kras* and loss of function in *Tp53* ($Kras^{G12D}Tp53^{-/-}$), whereas KL is defined by having oncogenic mutation in *Kras* and loss of function in a tumor suppressor gene *Stk11*, which encodes the protein LKB1 ($Kras^{G12D}Stk11^{-/-}$)[46]. NCI-H358, KP, and KL were

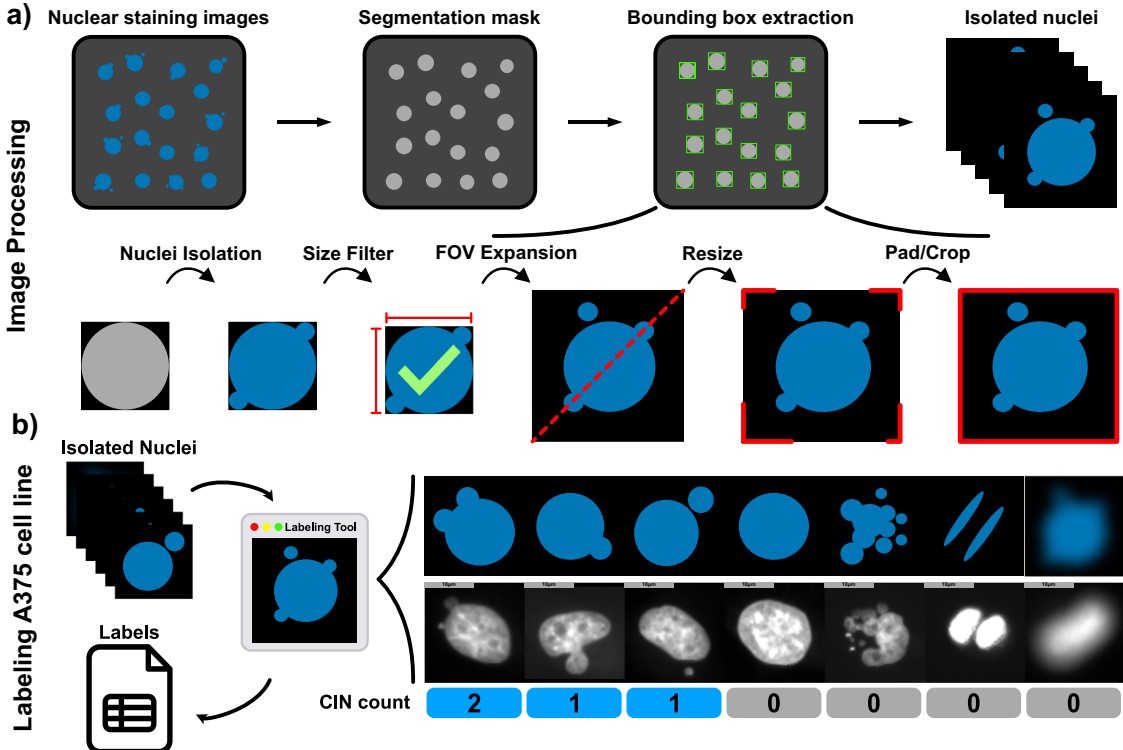

**Fig. 2 | Data preprocessing and labeling. a** micronuclAI image preprocessing initially requires the extraction of bounding boxes via a segmentation mask. Subsequently, Nuclei images are then processed by expanding the bounding box around each nucleus, removing small outliers, and resizing to 256 × 256 pixels while centering the object within the image. (Further details in the methods section). **b** With the help of a custom-made labeling tool, the CIN count of each isolated nuclei is recorded. The CIN count quantifies each MN and NBUD per isolated nuclei while specific patterns such as apoptotic nuclei, mitotic nuclei, and low quality or blurry images are given a CIN score of 0 to account for such structures present in real data. (All scale bars = 10 μm).

maintained in RPMI culture media supplemented with 10% FBS and 1% penicillin/streptomycin, whereas A375 were cultured in DMEM, supplemented with 10% FBS and 1% penicillin/streptomycin.

## Data Acquisition

Cells were seeded at a density of 1500 cells/well in opaque bottom 96-well tissue culture plates (Corning). Once confluency reached 70-80% in the well, cells were fixed in 4% paraformaldehyde for 15 min. After washing with PBS, cells were incubated with Hoescht (Thermo Fisher) diluted in Odyssey Blocking Buffer (1:10000). Fluorescent images were obtained on the Zeiss Celldiscover 7 using the PlanApochromat 20×/0.7 objective, 0.5× magnification changer, and Axiocam 506. Whole-well images were stitched and exported using Zen 3.1 Software resulting in a final resolution of 8829 x 9072 pixels.

## Segmentation

We extracted only the nuclear channel from raw images as ome.tif files using AICSImageIO (v 4.11.0)[47]. We performed nuclear segmentation on the resulting ome.tif images using: DeepCell Mesmer Whole-Cell (v 0.4.1), DeepCell Mesmer Nuclear (v 0.4.1), DeepCell Nuclear Segmentation (v 0.4.1)[34], Stardist (v 0.8.3)[36] and Cellpose (v 2.2.2)[35] on a high-performance computational cluster, bwHPC helix. DeepCell Mesmer and DeepCell Nuclear are deep-learning models used for cell segmentation and nuclear segmentation. DeepCell models are trained on a large number of images to achieve human level performance in the segmentation task. While DeepCell mesmer is trained on histological images, DeepCell Nuclear is optimized for segmentation on images derived from cell cultures. Cellpose is a human-in-the-loop generalist model for cell segmentation, providing some pre-trained models. In this case, we used the 'nuclei' model to perform nuclei segmentation. Stardist is a deep-learning based object detection with Star-convex shapes algorithm used for 2D and 3D object detection and segmentation in microscopy. Training data was

generated using Stardist generated segmentation masks and the corresponding nuclear ome.tif files.

## Nuclei isolation and preprocessing

Using the segmentation mask, we obtained the coordinates that circumscribe each of the segmented nuclei. We then use these coordinates to extract isolated nuclei (IN) patches from the nuclear images. To homogenize the apparent cell sizes of each single nuclei, we calculate a scaling factor for each one. The factor is calculated with the following formula:

$$Scaling\ Factor = \frac{Desired\ nuclei\ to\ image\ ratio \times max(Nuclei\ Width, Nuclei\ Height)}{Final\ Image\ size}$$

In this study, we use a nucleus to image ratio of 0.65; meaning each isolated nuclei occupies sixty five percent of the final image size. We increase the field of view (FOV) around each isolated nucleus by expanding the circumscribed bounding box (20 pixels in the case of the training data). To remove segmentation errors from the pipeline, we remove the bounding boxes falling into the first five percentile by area. The remaining isolated nuclei images are then resized and cropped to the center to a final resolution of 256 × 256 pixels. Lastly, we correct the brightness intensity in each image with brightness normalization. To facilitate this process, we developed the mask2bbox Python library, which is specifically designed for generating, handling, and visualizing bounding boxes from a segmentation mask image. Mask2bbox is available as a python package via the Python Package Index (PyPI), Bioconda, and as Docker and Singularity containers.

## Data labeling

A total of 84,286 isolated nuclei patches from the A375 cell line images at 10× were labeled by experts using a PySimpleGUI-based (v 4.60.4) labeling tool. This tool is available to use as a PyPI and Conda package. To label the

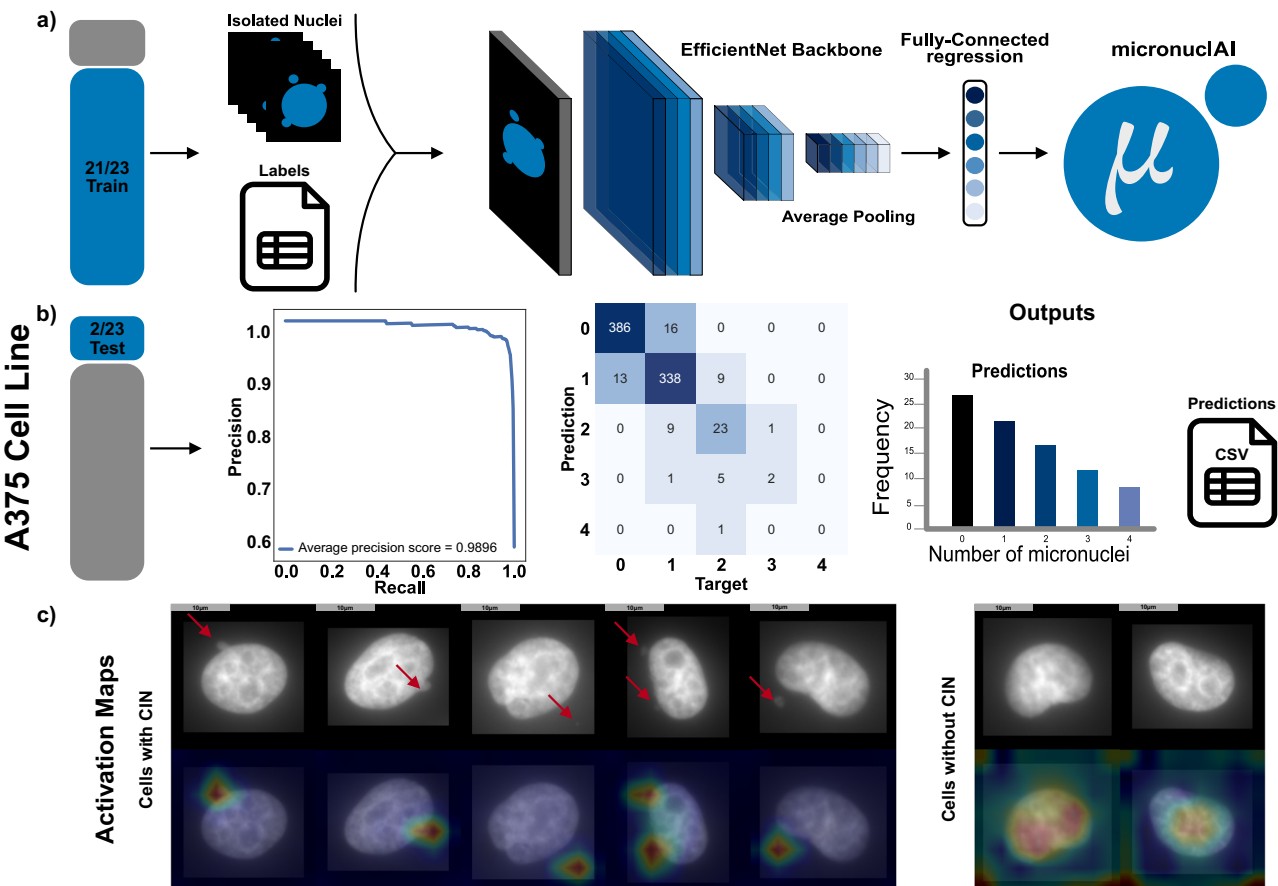

**Fig. 3 | Overview of the model training and testing. a** We train the micronuclAI model on the preprocessed and labeled isolated nuclei from DAPI stained images obtained from the A375 cell line. The training data consisted of 21/23 labeled images; the remaining 2/23 images were used as a hold out test dataset. We selected the best model based on the performance on the validation set. **b** The best model was then tested on a holdout test dataset consisting of 804 nuclei from 2 labeled images. With

an average precision score of 0.9860 micronuclAI offers high accuracy for the detection of CIN. During inference, micronuclAI outputs predicted CIN counts for each detected nuclei in a csv file along with significant statistics. **c** Activation maps show localized activation around MN in contrast to nuclei without CIN, where the activation is not localized and appears to diffuse around the primary nuclei. (All scale bars = 10 μm).

micronuclei, we adapted the protocol established by the HUMN project[42] for human lymphocytes, and the scoring criteria in oral exfoliated cells from[48] with some adjustments given the cell lines used and downstream application.

A micronuclei is counted if it fulfills the following criteria:
a. Rounded and smooth perimeter suggestive of a membrane, if separate from the primary nuclei.
b. Less than one-third of the diameter of the associated nucleus, but large enough to discern the shape.
c. Staining intensity similar to that of the primary nucleus.
d. Texture similar to that of the primary nucleus.
e. Same focal plane as the primary nucleus.

A NBUD is counted if it fulfills the above criteria for micronuclei and appears to be touching or budding out of the primary nuclei but is clearly distinguishable from/within the primary nuclear boundary.

Additionally, apoptotic and mitotic cells were identified with the following criteria and labeled as 0 CIN count.
a. Apoptotic cells: Single-cell patches with a large number of fragmented objects and no definitive primary nuclei were considered as apoptotic nuclei.
b. Mitotic cells: Single-cell patches with very bright double nuclei were considered as mitotic nuclei.

## Model Training
We used the labeled isolated nuclei images to train a convolutional neural network (CNN) to quantify the number of micronuclei associated with each

primary nucleus. The quantification is achieved using a model based on the EfficientNet[37] V2 architecture. The model architecture (Fig. 3a), is composed of a fully connected CNN, followed by a channel-wise convolution component as a feature extractor, and a fully connected layer for the final prediction. We employ versions of EfficientNet V1 and V2, which are models of increasing complexity, as the backbone feature extractors. The models were implemented in Pytorch (v 2.4.0)[49] using the Pytorch lightning framework (v 2.4.0)[49].

The models were trained using the mean squared error (MSE) as the loss function and Adaptive moment estimation (ADAM) as the gradient descent optimizer. The initial learning rate was set to 10e-3 and decayed using ReduceLRonPlateu (with a factor of 0.2 and a patience of 10 epochs) with an early stop criteria to a maximum of 300 epochs. For regularization, a dropout value of 0.2 was added to all fully connected (FC) layers. All EfficientNet (B0-B7) models were trained both with random initial weights and with weights of models pre-trained on ImageNet. Data augmentations were applied at random during training which include random vertical flip, random horizontal flip, random rotation ±0–30 degrees, randomly applied gaussian blur ($p = 0.3$, kernel_size = (3, 3)) and data normalization [−1, 1]. The mini-batch size was set to 64 for all models.

A 10-fold cross validation was performed over the entire dataset to compare between the different models. We evaluated and selected the backbone architecture based on the lowest root mean squared error (RMSE) value and F1-score on the validation set. Prediction values were rounded to the nearest integer for comparison with the training labels. Consequently,

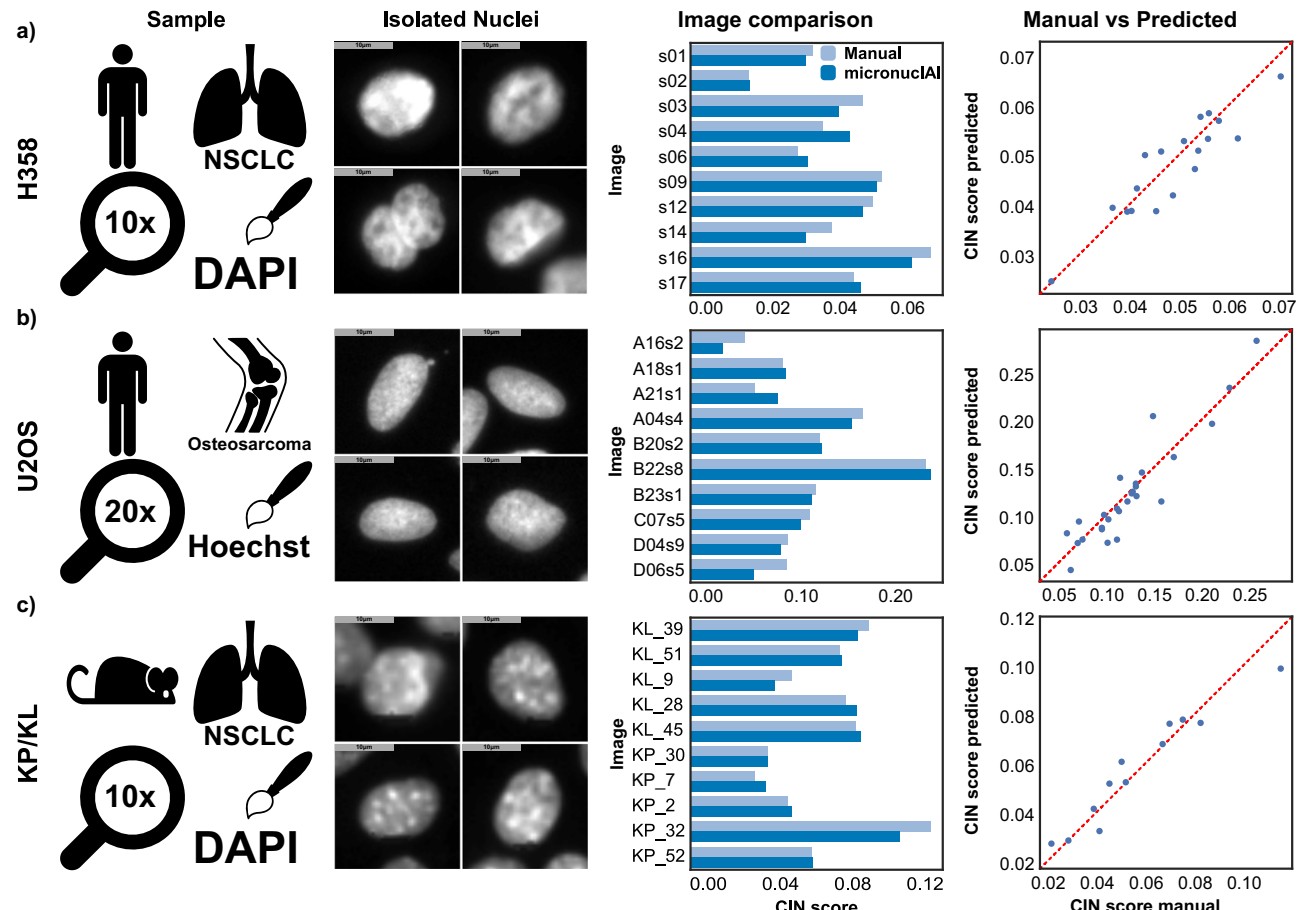

**Fig. 4 | micronuclAI pipeline validation results.** To validate the pipeline with a real-world application, we compared manual CIN scores against predictions obtained from micronuclAI for 4 different cell lines: **a** We used a DAPI-stained H358 human non-small-cell lung cancer derived cell line ($R^2$ = 0.8212, Person's Corr = 0.9124). **b** we used Hoechst stained U20S human osteosarcoma derived cell line obtained from the Broad Bioimage Benchmark Collection ($R^2$ = 0.8221, Pearson's Corr = 0.9278). **c** Lastly, we tested micronuclAI's capability to identify CIN on non-human cell lines by using the KP and KL cell lines derived from mouse non-small-cell lung cancer ($R^2$ = 0.9168, Pearson's Corr = 0.9620). For each cell line, we can observe the output of the nuclei isolation step resulting in homogeneous patches, which are then used as input for the model to make predictions on. (All scale bars = 10 μm).

**Table 2 | Results obtained on different cell types**

| Dataset | RMSE | Avg. Accuracy | R2 value | Pearson's Correlation |
|---|---|---|---|---|
| H358 | 0.00456 | 90.9269 | 0.8325 | 0.9124 |
| Broad u2os | 0.01436 | 86.6797 | 0.8609 | 0.9278 |
| KP/KL | 0.00684 | 91.1198 | 0.9194 | 0.9620 |
| Overall Average | | 89.5888 | 0.8709 | 0.9340 |

we trained the final model using the entire dataset to leverage all available data for potentially improved performance.

**Manual data annotation**

In addition to the manual labeling performed to generate the training data, we manually counted nuclei and MN on the cell lines which were used for validation of the complete pipeline. We used FIJI's (v 2.14.0 / 1.54f)[50] multi-point tool to count all nuclei present in an image while separately keeping track of all the MN and NBUDs. These values were then used to estimate the degree of CIN by calculating the CIN score, as a ratio between the total number of MN and total number of nuclei for each image. A Monte Carlo approach was used to estimate the CIN score in images where the number of cells was too large for manual labeling. This Monte Carlo estimation consisted of randomly selecting 3 FOV from each image and assessing MNs and

NBUDs for ~800–1000 nuclei per FOV. A summary of all the available data is present in Supplementary Table 3.

A Virtual Reality (VR) implementation was also developed that offers a gamified approach to annotation of single nuclei images. Since acquiring annotations is a laborious yet critical task, such an implementation may have great potential for increasing motivation from diverse communities within and outside of academia. A schematic of the VR application is presented in Supplementary Fig. 4. In brief, the VR environment reflects a 3D representation of the physical workspace, requiring linear scaling for object size and spacing to maintain spatial coherence (Fig. 4a). The key objects include the Patch Stack, a collection of individual cropped cell images stacked vertically; the Buckets, which act as containers where crops are assigned to indicate their micronuclei count (Fig. 4a); and the Whole Image, a display indicating the bounding box coordinates of a selected crop

(Fig. 4a). All objects are positioned symmetrically around the origin (O) at the center of the screen (Fig. 4a, b), with their size and spacing adjusted to fit within the respective field of view constraints (Fig. 4b). The patches can be moved around in 3D space and scaled up or down for better inspection. Upon session completion the recorded micronuclei counts are saved to a single tab separated text file that later is loaded as ground truth values for training the CNN model.

## Evaluation

**Evaluation of the CNN model.** We evaluated the CNN model on the hold-out test dataset within the initial isolated nuclei images. The model prediction values for the CIN count associated with each isolated nuclei were rounded to the closest integer value. The model performance was then assessed with standard classification metric, F1-score and Precision-Recall curve using the scikit-learn implementation in Python (v 1.0.2)[51]. We also generated attention maps with Class Activation Maps (CAM)[52] as a part of our attempt at generating explainable AI (XAI). Attention maps provide a visual representation of how the neural network processed the information helping us understand and confirm the model's decisions.

**Evaluation of the pipeline on the H358 cell-line.** We evaluated the performance of the pipeline on images taken from the H358 cell-line, a primary bronchoalveolar carcinoma of the lung, a non-small-cell lung cancer. The dataset contains 17 DAPI-stained images, 10k × 10k pixels imaged at 10× containing approximately 20,000 nuclei in total. This set of images consisted of 3 different genetic variations that alter the CIN level : H358, H358-GFP, H358-dnMCAK in order of CIN level from low to high. The evaluation was done on the basis of CIN score for each image compared to manual expert quantifications. $R^2$ and Pearson's correlation were calculated as a summary statistic.

**Evaluation of the pipeline on an external dataset imaged at 20×.** To test the scale invariability of the method, we evaluated the performance of micronuclAI with an external image dataset BBBC039v1[39], available from the Broad Bioimage Benchmark Collection[53]. The dataset contained Hoechst-stained human U2OS cells with 200 fields of view, 520 × 296 pixels imaged at a 20× zoom using ImageXpress Micro epifluorescent microscope (Molecular Devices). Evaluation was done on a set of 20 random images from this dataset containing approximately 2400 nuclei by comparing the manual and micronuclAI estimated CIN score using both $R^2$ and Pearson correlation as summary statistics.

**Evaluation of the pipeline on murine cell lines.** We further tested micronuclAI on mice derived KP/KL cell lines: KP ($Kras^{G12C}Tp53^{-/-}$) and KL ($Kras^{G12C}Lkb1^{-/-}$). 5 KP and 5 KL images were manually annotated and compared to micronuclAI predictions. Given the large number of cells images (~20 K per image) we used the Monte Carlo estimation counting strategy for this task.

## Implementation / Deployment

The complete micronuclAI pipeline is available as a Command Line Interface (CLI) through Github, a simple Graphical User Interface (GUI) through Streamlit, and a nf-core[54] compliant pipeline for high-throughput image analysis. Inference of micronuclei can be achieved in small to medium sized example TIFF/OME-TIFF images that can be uploaded to the streamlit app. Image data is processed within a virtual machine (VM) on Heicloud, a local Cloud infrastructure provided by University Computing Center Heidelberg, and images are immediately deleted after micronuclei inference. Once micronuclei are inferred, results predictions as well as several plots describing the results are generated and presented to the user within the Streamlit app. No data is kept on the server after the user disconnects from the app.

## Statistics and reproducibility

Statistical analyses were performed with Python version 3.11. The datasets used for evaluation in this study contained 17, 20, and 10 images for the H358, BBC039v1, and KP/KL datasets respectively. Each image with the same modification was defined as a technical replica for measuring the overall micronuclei ratio. The sample size for each experiment including the number of images used and the number of cells was explained in the Methods section and figure legend.

## Data availability

Images, segmentation masks and labels used during model training are available at synapse under the link https://www.synapse.org/#!Synapse: syn54780485/. One of the test datasets BBBC039v1, obtained from the Broad Bioimage Benchmark Collection can be found at https://bbbc. broadinstitute.org/. The numerical source data used for the statistical analysis and figure generation is available in "Supplementary Data".

## Code availability

The micronuclAI training and inference code is available through Github at https://github.com/SchapiroLabor/micronuclAI and the current version is deposited at Zenodo[55]. The Labeling Tool code is available through Github at https://github.com/SchapiroLabor/micronuclAI_labeling and the current version is deposited at Zenodo[56]. The code for the micronuclAI Streamlit App is available through Github at https://github.com/SchapiroLabor/micronuclAI_streamlit and the current version is deposited at Zenodo[57]. The micronuclAI Nextflow Pipeline is available through Github at https://github.com/SchapiroLabor/micronuclAI_nf and the current version is deposited at Zenodo[58]. The VR Labeling tool is available through Github at https://github.com/SchapiroLabor/micronuclAI-VR and the current version is deposited at Zenodo[59]. The Mask2Bbox package developed for the tool is available through Github at https://github.com/SchapiroLabor/mask2bbox and the current version is deposited at Zenodo[60].

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

## Acknowledgements

The authors acknowledge support by the state of Baden-Württemberg through bwHPC and the German Research Foundation (DFG) through grant INST 35/1597-1 FUGG. A.H. was supported through state funds approved by the State Parliament of Baden-Württemberg for the Innovation Campus Health + Life Science Alliance Heidelberg Mannheim. D.S., M.A.I.A., N.S., and F.W. was supported by the German Federal Ministry of Education and Research (BMBF 01ZZ2004). F.W. was supported by a Walter-Benjamin position from the Deutsche Forschungsgemeinschaft (DFG). B.I. is supported by National Institute of Health grants, R37CA258829, R01CA280414, R01CA266446, U54CA274506, and additionally by the Pershing Square Sohn Cancer Research Alliance Award, the Burroughs Wellcome Fund Career Award for Medical Scientists; a Tara Miller Melanoma Research Alliance Young Investigator Award; the Louis V. Gerstner, Jr. Scholars Program; and the V Foundation Scholars Award. Benjamin Izar, MD, PhD is a CRI Lloyd J. Old STAR (CRI5579). L.A.C is supported by NIH/NCI grant F30CA281104-01 and, along with Z.H.W. and L.C., MSTP Training Grant T32GM007367. This work is supported by the Health + Life Science Alliance Heidelberg Mannheim and received state funds approved by the State Parliament of Baden-Württemberg. ("AI Health Innovation Cluster" and "MULTI-SPACE"). For the publication fee we acknowledge financial support by Heidelberg University. We thank our administrative and project management team: Erika Schulz, Lydia Roeder and Bettina Haase. We would also like to thank Ricardo Omar Ramirez Flores, Jovan Tanevski, Victor Perez, and Chiara Schiller for the helpful discussions and Cristina-Ruxandra Burghelea for her help in proofreading the manuscript.

## Author contributions

M.A.I.A.: Methodology, Interpretation, Writing, Figures, Nextflow Pipeline. L.A.C.: Data Acquisition, Methodology, Interpretation, Writing. A.H.: Methodology, Interpretation, Writing. F.W: Streamlit app. N.S.: Methodology. L.C.: Data Acquisition. S.S.: Data acquisition. Z.H.W. Data acquisition. J.C.M.: Data acquisition, Methodology. K.B.: Nextflow Pipeline. I.M.: VR Labeling tool. B.I.: Conceptualization, Interpretation, Writing, Supervision, Funding. D.S: Conceptualization, Interpretation, Writing, Supervision, Funding.

## Funding

## Competing interests

D.S. reports funding from Cellzome, a GSK company and received honorariums from Immunai, Noetik, Alpenglow and Lunaphore. B.I. is a consultant for or received honoraria from Volastra Therapeutics, Johnson & Johnson/Janssen, Novartis, Eisai, AstraZeneca and Merck, and has received research funding to Columbia University from Agenus, Alkermes, Arcus Biosciences, Checkmate Pharmaceuticals, Compugen, Immunocore, Regeneron, and Synthekine. All other authors declare no competing interests.
