## [Transparent Peer Review file · Communications Biology]

micronuclAI enables automated quantification of micronuclei for assessment of chromosomal instability.

Corresponding Author: Dr Denis Schapiro

Version 0:

Reviewer comments:

Reviewer #1

(Remarks to the Author)

Micronuclei formation is a standard method for assessing chromosomal instability in cell lines and in tissue samples. In order to get a reliable assessment for micronuclei prevalence, it is necessary to laboriously image and score hundreds or thousands of cells.

In this paper, Ibarra-Arellano et al. describe the new tool they have designed to circumvent this. They propose that their micronuclAI software can take over this work by automatically scoring images. The group use samples from two human cell lines, as well as a third where a construct was used to exacerbate CIN, as well as a cultured mouse cell line. They use different DNA-staining chemicals such as DAPI and Hoescht, and also image cells using different levels of magnification. In all examples, their automated tool scores micronuclei prevalence at similar levels as manual counting, suggesting it is working reliably. It can also count for number of MN per cell. This suggests the tool will be a useful for the field.

I only have minor comments. The name is very cute, but could be confusing when being used whether one is talking about the tool or actual micronuclei. Secondly, there was no mention of tissue samples with H&E staining, is this something that the tool can handle? Also it would be helpful to show example images of where the tool gave misleading answers, if possible, so that the limitations can be assessed. For example, what happens if the cells are prone to growing on top of each other? Finally, have they tested images from another microscope? What image formats will the software accept?

Reviewer #2

(Remarks to the Author)

Overview

This study presents a method, micronuclAI, which can detect micronuclei (MN) in DNA-only stained images as an indicator of chromosomal instability (CIN). micronuclAI can accurately predict micronuclei numbers in multiple cell lines and species with comparable performance to human. They also present an online platform where micronuclAI can be utilized to make micronuclei predictions.

The objectives and rationale of the study are clearly stated, the methods are fairly described and provided to ensure replicability and/or reproducibility. The statistical analysis and reporting are appropriate but can be more clearly described (see below for details).

The current tables, figures and information flow is satisfactory. The interpretation of the results and conclusions are generally supported by the data. Also, the authors clearly emphasized the strengths as well as the limitations. In addition, the corresponding GUI was developed and deployed which would help increase the accessibility of the current study. The reviewer has some concerns regarding the robustness in methodology used, such as the lack of method performance testing and use of blurry images.

Major comments

1. Lack of cross-validation based testing: Why do the authors use only one train/test split as opposed to k-fold cross validation, which is usually a more robust way to train the data? This may provide a more robust model as only 21 whole-images are used and many nuclei from one image may lead to overfitting. Please perform this, considering Point 2 in mind.
2. Are the segments from the same image stratified evenly in training/validation/test folds? If not, this should be conducted to improve model robustness. This is unclear.
3. We request the authors to add a section focusing on analyzing cases where the model does not do well. Such analysis and its description will allow the users to know the fail-cases and how to use the tool better.
4. While we understand the basis of the approach of down-sampling negative MN cases to match the number of positive samples, it is possible that model may learn even better without this down-sampling. Thus, we encourage the authors to try the model using all the data and present the results. While a large number of single-cell images were used during training, it only comprised a total of 21 whole-images, and keeping the large number of negative samples may provide further variance to improve generalizability.
5. Are there any procedures to assess the quality of the images or segments? Why are blurry images given a value of zero? It seems, to improve model performance, clear images should be prioritized when conducted down-sampling (with the inclusion of some blurry images to replicate real-life scenarios). This is especially true because of the significant down-sampling that is being conducted.
6. The authors did take steps to address potential issues related to patch overlap and double counting - removing small patches and calculate IoU, however, the authors did not explicitly state any additional steps to account for the potential homogeneity of patches from the same image versus different images.
 - Ideally, it should be ensured that patches from the same image do not dominate the training process. (Same as point 2)
 - Further checkups are recommended to ensure that there is no data leakage during training.

Minor Comments

7. Line 121-122: I suggest adding elaboration of the mask selection criterion/process, what does "most accurate results" mean, what criteria were used?
8. Line 149: the number of the images used to isolate the 1,306 nuclei was not specified.
9. Line 163-165: In figure 3c, it is a bit unclear which are the ones with CIN and which are the ones without CIN. According to the current caption, seems like the left 5 are the ones with CIN but why are the activation areas towards the border of the image, which are relatively less informative?
10. Line 275: I suggest adding formal description of the source of KP and KL cell line instead using another researcher's name.
11. Line 288 & reference 45: ALCSImageIO source (URL) is not well embedded, cannot be used to track the original webpage at this point.
12. Line 314: I suggest adding elaboration on the resize process of the images.
13. Line 350: I suggest adding reference record for EfficientNet.
14. Figure 1c resolution can be improved.

Version 1:

Reviewer comments:

Reviewer #1

(Remarks to the Author)

I thank the authors for addressing my comments. If they can add a line to the manuscript, saying it is not suitable for H&E stained images but limited to DAPI-stained cell line samples etc, then I am happy with the revision.

Reviewer #2

(Remarks to the Author)

I got a chance to look at this and found the replies to our comments, respective analysis and additions to the manuscript very satisfactory. They were able to address the needed gaps in cross validation, data leakage issues, handling imbalanced cases and thus building a software for micronuclei detection that can be very useful to the field.

I recommend accepting the paper and I have no further comments.

Thanks!

We thank the reviewers for their thoughtful review of our manuscript. They raised important comments about the way we trained micronuclAI and provided useful feedback to improve upon it. We believe that addressing these comments has led to a much improved manuscript and tool. Below, we provide the responses to the individual comments and outline the changes that were made.

Reviewers' comments:

Reviewer #1 (Remarks to the Author):

Micronuclei formation is a standard method for assessing chromosomal instability in cell lines and in tissue samples. In order to get a reliable assessment for micronuclei prevalence, it is necessary to laboriously image and score hundreds or thousands of cells.

In this paper, Ibarra-Arellano et al. describe the new tool they have designed to circumvent this. They propose that their micronuclAI software can take over this work by automatically scoring images. The group use samples from two human cell lines, as well as a third where a construct was used to exacerbate CIN, as well as a cultured mouse cell line. They use different DNA-staining chemicals such as DAPI and Hoescht, and also image cells using different levels of magnification. In all examples, their automated tool scores micronuclei prevalence at similar levels as manual counting, suggesting it is working reliably. It can also count for number of MN per cell. This suggests the tool will be a useful for the field.

Author response: Thank you! We are grateful to the reviewer's comments regarding the usefulness of micronuclAI. We believe the tool will provide researchers a faster, and reliable method to process large amounts of data for assessment of Chromosomal Instability.

Comment 1

I only have minor comments. The name is very cute, but could be confusing when being used whether one is talking about the tool or actual micronuclei.

Author response: We acknowledge the reviewer's comments regarding the name of the tool but we would like to request the reviewer to keep the current name for the tool since we believe the two words in terms of pronunciation are sufficiently different so as to not confuse between them, micronuclei and micronuclAI (with a more emphasis on the AI part).

Comment 2

Secondly, there was no mention of tissue samples with H&E staining, is this something that the tool can handle?

Author response: We appreciate the reviewer's suggestion regarding the potential for micronuclAI to analyze CIN in tissue samples with H&E staining. Quantifying CIN in H&E-stained tissue samples, however, presents substantial challenges. CIN is frequently generated due to errors in cellular division, and thus the accurate quantification of this type of CIN (referred to as numerical chromosomal instability) necessarily depends on optimization of cell culture conditions.

Factors such as the timing of fixation and the initial seeding density of cells are controlled and thus allow for a substantial number of mitotic events to be captured and thus a more precise assessment of CIN *in vitro*. As highlighted in this manuscript, this transfers well to other experimental setups involving cell culture conditions.

In contrast, tissue samples do not allow the same degree of control over these variables, which can impact CIN assessment accuracy. For example, CIN-high tissues may not display micronuclei at the same frequency observed in cultured cells (e.g., a 25% MN rate in cells may not be reached in tissues). Furthermore, the recovery of micronuclei, in addition to mitotic events in H&E images, is infrequent and is difficult to assess in H&E images. Technical issues including but not limited to the presence of artifacts and the variability of tissue appearance, in addition to biological factors such as the presence of overlapping or irregularly shaped cells (a difficulty encountered by the current tool), and the presence of “other” nuclei (i.e. local lymphocytes, fibroblast populations in the tumor microenvironment) make segmentation and manual annotation of H&E images difficult for even experienced pathologists, and would need to be accounted for in order to train an AI-guided model.¹

A more promising approach to CIN quantification in tissues would involve fluorescent staining (instead of H&E staining) using e.g., cGAS as an indicator of micronuclei, which could be assessed with micronuclAI. We currently have a cohort of cGAS-stained brain metastasis tissue images from a previous study, which may serve as a valuable training dataset for the model to refine its capacity for CIN evaluation in tissue samples. However, at this point, this extension is out of scope for this manuscript.

Comment 3

Also it would be helpful to show example images of where the tool gave misleading answers, if possible, so that the limitations can be assessed. For example, what happens if the cells are prone to growing on top of each other?

Author response: We thank the Reviewer for this insightful comment and we now present exemplary data in Supplementary Figure 3 of the revised manuscript. We summarize the findings below:

Cases where the model does not perform well are mostly attributed to instances in images with overlapping nuclei (due to cell density), irregularly shaped nuclei, or the presence of apoptotic cell fragments in the periphery of isolated nuclei. Furthermore, instances of the same MN on two different isolated nuclei images might be present, leading to a potential double counting of the same structures within two different patches. Such instances occur on areas of the image with high nuclear/cell density that might yield misleading or inflated results.

Supplementary Figure 3: Examples of edge cases where the model may not perform well

Figure S3: **micronuclei detection challenges in overlapping and overcrowded cell environments.** **a**, Overlapping cells growing on top of each other can sometimes appear to contain micronuclei (MN), as their nuclei are often segmented as a single entity. In such cases, micronucleiAI typically provides a score of 0 MN; however, in some instances, additional nuclei are misinterpreted as nuclear buds, resulting in a score of 1 MN. **b**, Overcrowding in the field of view (FOV), where nuclei are surrounded by neighboring nuclei or fragments, can lead to misidentification, particularly when adjacent nuclei are undergoing e.g., apoptosis. This disintegration can mimic the appearance of multiple micronuclei, complicating accurate MN identification.

Comment 4

Finally, have they tested images from another microscope? What image formats will the software accept?

Author response: We thank the reviewer’s comment pointing at the lack of this information in the manuscript. Yes, we have evaluated micronucleiAI with three different datasets encompassing different nuclei staining, organisms, objectives and microscopes. While the cell cultures A375, KP/KL, and H358 were obtained using the ZEISS Celldiscoverer 7s, the external dataset from the Broad Institute was captured at 20x magnification on an ImageXpress Micro epifluorescent microscope (Molecular Devices)². Regarding the image format, the currently accepted input file formats are (OME)-TIFF but could be extended to other bio-formats compatible file formats.

We have added the following sentences in the manuscript:

Line 423-425: “The dataset contained Hoechst-stained human U2OS cells with 200 fields of view, 520x296 pixels imaged at a 20x magnification using ImageXpress Micro epifluorescent microscope (Molecular Devices).”

Line 436-438: “Inference of micronuclei can be achieved in small to medium sized example (OME)-TIFF images that can be uploaded to the streamlit app. Image data is processed within a virtual machine (VM) on Heicloud”

Reviewer #2 (Remarks to the Author):

Overview

This study presents a method, micronuclAI, which can detect micronuclei (MN) in DNA-only stained images as an indicator of chromosomal instability (CIN). micronuclAI can accurately predict micronuclei numbers in multiple cell lines and species with comparable performance to human. They also present an online platform where micronuclAI can be utilized to make micronuclei predictions.

The objectives and rationale of the study are clearly stated, the methods are fairly described and provided to ensure replicability and/or reproducibility. The statistical analysis and reporting are appropriate but can be more clearly described (see below for details).

The current tables, figures and information flow is satisfactory. The interpretation of the results and conclusions are generally supported by the data. Also, the authors clearly emphasized the strengths as well as the limitations. In addition, the corresponding GUI was developed and deployed which would help increase the accessibility of the current study. The reviewer has some concerns regarding the robustness in methodology used, such as the lack of method performance testing and use of blurry images.

Author response: We thank the reviewer for the positive comments, insightful questions, and suggestions. These have helped make our methodology and statistical analysis much clearer and improved the implementation of the overall tool.

Major Comments

Comment 1: Lack of cross-validation based testing: Why do the authors use only one train/test split as opposed to k-fold cross validation, which is usually a more robust way to train the data? This may provide a more robust model as only 21 whole-images are used and many nuclei from one image may lead to overfitting. Please perform this, considering Point 2 in mind.

Author response: Thank you for the comment. We agree that a k-fold cross validation results would display the robustness of the model better. Therefore, we performed a 10-fold CV where the 10-fold models had similar performance suggesting the model’s performance is stable across different data splits. Taking the reviewer’s comment into account, we have modified the following sections in the Methodology and Results section:

Methodology Section: Line 369 - 373

“A 10-fold cross validation was performed over the entire dataset to compare between the different models. We evaluated and selected the backbone architecture based on the lowest root mean squared error (RMSE) value and F1-score on the validation set. Prediction values were rounded to the nearest integer for comparison with the training labels. Consequently, we trained the final model using the entire dataset to leverage all available data for potentially improved performance.”

Results/Discussion Section: Line 130-132

“We selected the EfficientNet-V2-S model architecture as the final model based on the average 10-fold cross-validation (CV) performance. The consistent performance across all the folds validated this decision, ensuring that our model is both robust and well-generalized.”

For the best model (EfficientNet V2 - S), we also include a table below showcasing the results obtained for the 10-fold CV.

Fold	F1	RMSE
0	0.9293	0.2835
1	0.9452	0.2549
2	0.9419	0.2733
3	0.9203	0.2961
4	0.9219	0.3017
5	0.9408	0.2705
6	0.9294	0.2808
7	0.9155	0.3097
8	0.9253	0.2921
9	0.9404	0.2689
Average	0.931	0.28315

Comment 2: Are the segments from the same image stratified evenly in training/validation/test folds? If not, this should be conducted to improve model robustness. This is unclear.

Comment 6: The authors did take steps to address potential issues related to patch overlap and double counting - removing small patches and calculating IoU, however, the authors did not explicitly state any additional steps to account for the potential homogeneity of patches from the same image versus different images.

- Ideally, it should be ensured that patches from the same image do not dominate the training process. (Same as point 2).
- Further checkups are recommended to ensure that there is no data leakage during training.

Author response:

We are thankful to the reviewer for highlighting these points. These comments made us realize that we could have had data leakage (model might have seen the data in the test set) since the training/validation/test came from a mixed pool of single nuclei patches obtained from all the images.

Based on this feedback, we have changed our approach by using 2 hold-out images that were not present in the train/validation split for testing the model, this guarantees no data leakage during training. Using this approach, we did not require an even stratification for training/validation/test folds but rather a stratification by label during training to ensure class balance. We included this information into the methods section.

Methods: Line 121 - 128

“To better handle this data imbalance, we removed 23 (0.0272%) outliers with a CIN count ≥ 4 , and randomly sampled the same number of nuclei with a CIN count of 0 to the nuclei with a CIN count > 0 . The balanced training dataset consisted of 12,304 nuclei from 21 of the 23 labeled images; from which 6,152 (50%) had CIN count of 0; 5,473 (42.74%) had a CIN count of 1; 564 (4.58%) had a CIN count of 2; and 92 (0.75%) had CIN count of 3 (Fig. 2b). From this balanced dataset of 12,304 nuclei, (90%) were used for training and (10%) used for validation. The test dataset consisted of 804 nuclei obtained from the remaining 2 hold out test images (Fig. 3a).”

To further check if there is a large disparity of the frequency of patches coming from the same image, we plotted the distribution of patches coming from each image. The number of individual single cell patches obtained from different whole images were found to be more or less similar. This ensures that nuclei patches that come from any single whole image do not dominate the training process.

Comment 3: *We request the authors to add a section focusing on analyzing cases where the model does not do well. Such analysis and its description will allow the users to know the fail-cases and how to use the tool better.*

Author response:

We thank the reviewer for this valuable suggestion. We agree that understanding the limitations of the model is crucial for our users. In our study, we observed that the model performed consistently well across most test cases, making it challenging to identify a substantial number of clear failure cases. However, we acknowledge that this could be due to limitations in our current testing scenarios or imaged cell lines.

We now present exemplary data highlighting the Reviewer’s suggestion in Supplementary Figure 3 in the revised manuscript. We summarize the findings below:

Cases where the model does not perform well are mostly attributed to instances in images with overlapping nuclei (due to cell density), irregularly shaped nuclei, or the presence of apoptotic cell fragments in the periphery of isolated nuclei. Furthermore, instances of the same MN on two different isolated nuclei images might be present, leading to a potential double counting of the

same structures within two different patches. Such instances occur on areas of the image with high nuclear/cell density that might yield misleading or inflated results.

Supplementary Figure 3: Examples of edge cases where the model may not perform well

Figure S3: **micronuclei detection challenges in overlapping and overcrowded cell environments.** **a**, Overlapping cells growing on top of each other can sometimes appear to contain micronuclei (MN), as their nuclei are often segmented as a single entity. In such cases, micronuclei typically provides a score of 0 MN; however, in some instances, additional nuclei are misinterpreted as nuclear buds, resulting in a score of 1 MN. **b**, Overcrowding in the field of view (FOV), where nuclei are surrounded by neighboring nuclei or fragments, can lead to misidentification, particularly when adjacent nuclei are undergoing e.g., apoptosis. This disintegration can mimic the appearance of multiple micronuclei, complicating accurate MN identification.

Comment 4: *While we understand the basis of the approach of down-sampling negative MN cases to match the number of positive samples, it is possible that the model may learn even better without this down-sampling. Thus, we encourage the authors to try the model using all the data and present the results. While a large number of single-cell images were used during training, it only comprised a total of 21 whole-images, and keeping the large number of negative samples may provide further variance to improve generalizability.*

Author response:

Yes, we agree with the reviewer that it is possible that the model may learn even better without this down-sampling. We therefore ran multiple experiments with all the data present without any significant down-sampling of the majority class (i.e. Unbalanced case) and with down-sampling to balance the class distribution (i.e. Balanced case). We present the findings with the following table:

Down-sampling	Base Model	K-Folds	Size	Mean-F1	Mean- RMSE
Balanced	EfficientNet V1 - B0	10	256	0.9186	0.30573
Unbalanced	EfficientNet V1 - B0	10	256	0.901	0.34259
Balanced	EfficientNet V1 - B0	10	128	0.9147	0.3178
Unbalanced	EfficientNet V1 - B0	10	128	0.8954	0.3482
Balanced	EfficientNet V2 - S	10	256	0.9215	0.3008
Unbalanced	EfficientNet V2 - S	10	256	0.8989	0.3561
Balanced	EfficientNet V2 - S	10	128	0.9229	0.3008
Unbalanced	EfficientNet V2 - S	10	128	0.8954	0.3483

The models trained on the balanced dataset outperformed the models trained on the unbalanced dataset. We believe the model performance for the unbalanced dataset decreased as the training dataset is severely skewed to the majority class as in real-world examples and would hence misclassify the limiting classes. Additionally, we observe a consistent increase in performance on the models trained using the EfficientNet V2 - S architecture.

Results

The following text has been added to the results section line 132 - 135

“Furthermore, we compared the performance of the model using both a balanced and an imbalanced dataset. This comparison highlighted better model performance when trained on the balanced dataset (Supp. table 2).”

Comment 5: *Are there any procedures to assess the quality of the images or segments? Why are blurry images given a value of zero? It seems, to improve model performance, clear images should be prioritized when conducting down-sampling (with the inclusion of some blurry images to replicate real-life scenarios). This is especially true because of the significant down-sampling that is being conducted.*

Author response:

We acknowledge that prioritizing clear images is crucial for enhancing model performance, particularly when significant down-sampling is conducted.

We assessed the quality of the whole images and individual segments visually through manual inspection. Whole images with a large number of out of focus nuclei or regions were discarded. Only images with an appropriate level of clear ROIs were selected resulting in 23 images used to train and test the model. This approach would limit the number of blurred nuclei in the complete dataset even before we down-sample.

In order to make this process quantifiable and automatic, we have implemented a simple blur detection algorithm using the gradient of Laplacian on the single cell crops. The number of blurry nuclei crops present in the training data was then quantified and found to be 10%. Furthermore, we also include the amount/percentage of blurry cells as a QC metric in the output.

The blurry images were given a value of zero to remain true to the manual quantification method where blurry nuclei must still be quantified. While the number of micronuclei cannot be quantified from such images, imaging experiments often result in such out of focus regions within the image which need to be quantified.

We additionally trained models using clean data (i.e. where there are no blurry nuclei present) and data limited to 10% of blurry crops to see the impact on model performance. Here, we observed similar trends for balanced and unbalanced dataset and comparable F1 and RMSE scores to the unfiltered training data. This shows that inclusion of around 10% blurry nuclei does not affect performance and may be better for real-world scenarios where such cases can be common.

Downsampling	Data QC	Base Model	K-Folds	Size	F1	RMSE
Balanced	Clean	EfficientNet V2 - S	10	128	0.9225	0.2997
Unbalanced	Clean	EfficientNet V2 - S	10	128	0.9004	0.3410
Balanced	10%	EfficientNet V2 - S	10	128	0.9186	0.3106
Unbalanced	10%	EfficientNet V2 - S	10	128	0.8990	0.3425

In this regard, we have added the following text to the Discussion part of the manuscript Line 244-247:

“Furthermore, our pipeline is also robust against images of out-of-focus nuclei, which are common in microscopy images (Supplementary Table 3). The pipeline outputs a percentage of out-of-focus nuclei as a form of quality check. So far, the pipeline has been tested with inclusion of around 10% blurry nuclei without a negative impact on the performance.”

Minor Comments

Comment 7: Line 121-122: *I suggest adding elaboration of the mask selection criterion/process, what does “most accurate results” mean, what criteria were used?*

Author response

To clarify this, we have changed the sentence in line 110-112 to “We visually inspected the generated nuclear segmentation masks and chose Stardist which generated the most accurate masks through qualitative comparison of several ROIs within the input image”.

Comment 8: Line 149: *the number of the images used to isolate the 1,306 nuclei was not specified.*

Author response:

Thank you for pointing this out. Since we changed the way we are training and testing the model, in order to avoid data leakage, the current number of nuclei used for testing changed to 804. These 804 nuclei come from two images that were not used for training. You can see the change reflected in the manuscript line 126 -128.

“From this balanced dataset of 12,304 nuclei, (90%) were used for training and (10%) used for validation. The test dataset consisted of 804 nuclei obtained from the remaining 2 hold out test images (Fig. 3a)”

Comment 9: Line 163-165: *In figure 3c, it is a bit unclear which are the ones with CIN and which are the ones without CIN. According to the current caption, it seems like the left 5 are the ones with CIN but why are the activation areas towards the border of the image, which are relatively less informative?*

Author response:

There is no definitive answer to why the activation areas are towards the borders of the image without any degree of CIN. Visually to us, it looks as if the activation is more diffused over the whole image as the network focuses over the complete image when it does not find any micronuclei close to the primary nuclei.

The figure has been updated to make this clearer.

Comment 10: Line 275: *I suggest adding a formal description of the source of KP and KL cell lines instead using another researcher’s name.*

Author response:

Thank you for pointing this out. We have added formal descriptions for both KP and KL cell lines in the manuscript on lines 275-279:

“KP (KrasG12DTp53^{-/-}) and KL (KrasG12DStk11^{-/-}) are descriptors of a syngeneic pair of murine non-small cell lung cancer lines. KP is characterized by both oncogenic mutations in Kras and loss of function in Tp53 (KrasG12DTp53^{-/-}), whereas KL is defined by having oncogenic mutation in Kras and loss of function in a tumor suppressor gene called Stk11, which encodes the protein LKB1 (KrasG12DStk11^{-/-})”

Comment 11: Line 288 & reference 45: *ALCSImageIO source (URL) is not well embedded, cannot be used to track the original webpage at this point.*

Author response:

The URL link has now been properly embedded. AICSImageIO was cited as requested by the authors in their documentation linked here.

Comment 12: Line 314: *I suggest adding elaboration on the resize process of the images.*

Author response:

The caption for Figure 2 at line 138-140 has been changed from:

“The isolated nuclei are subjected to various preprocessing steps to capture the immediate surrounding and make them homogeneous in size.”

to:

“Nuclei images are then processed by expanding the bounding box around each nucleus, removing small outliers, and resizing to 256x256 pixels while centering the object within the image. (Further details in the methods section.)”

Additionally, we now include the algorithm in the methods section of the manuscript.

Algorithm 1 Preprocessing of input images

- 1: Isolate the single nuclei crops images with the help of the nuclear masks
 - 2: Calculate the scaling factor (s) with a desired nuclei to image ratio of 0.60
 - 3: Expand the FOV around the nuclei by n pixels (default: n = 25)
 - 4: Remove objects falling into the first five percentile by area
 - 5: Resize the final image as the object in the center to 256x256
-

Comment 13: Line 350: *I suggest adding a reference record for EfficientNet.*

Author response:

We have added a reference for EfficientNet.

Comment 14: Figure 1c resolution can be improved.

Author response:

Thank you for the comment. We have changed the figure to showcase a mock-up of the web application instead of a screenshot of it.

References

1. Veta, M., Pluim, J. P. W., van Diest, P. J. & Viergever, M. A. Breast cancer histopathology image analysis: a review. *IEEE Trans. Biomed. Eng.* **61**, 1400–1411 (2014).
2. Gustafsdottir, S. M. *et al.* Multiplex Cytological Profiling Assay to Measure Diverse Cellular States. *PLoS One* **8**, e80999 (2013).

Response to Referees

Reviewer #1 (Remarks to the Author):

I thank the authors for addressing my comments. If they can add a line to the manuscript, saying it is not suitable for H&E stained images but limited to DAPI-stained cell line samples etc, then I am happy with the revision.

We thank reviewer #1 for all the great suggestions. We have added the following sentences to highlight the limitations of the method.

line 98-102: micronuclAI distinguishes itself over previous methods 26–32 as it 1) can quantify for both MN and NBUDs; 2) requires only nuclear (DNA) staining; 3) is able to work with 10x to 20x image objectives; 4) can work with any segmentation mask, and most importantly 5) is extensively evaluated in multiple cell lines and thus, ready for use by the community **for nuclear stained images of cell lines** (Fig. 1 and Table 1).

Line237-239: **Lastly, we would like to note, that H&E staining as performed in routine diagnostic is currently not supported due to missing ground-truth data.**

Reviewer #2 (Remarks to the Author):

I got a chance to look at this and found the replies to our comments, respective analysis and additions to the manuscript very satisfactory. They were able to address the needed gaps in cross validation, data leakage issues, handling imbalanced cases and thus building a software for micronuclei detection that can be very useful to the field.

I recommend accepting the paper and I have no further comments.

We thank reviewer #2 for all the great suggestions.